# Pretherapeutic Imaging for Axillary Staging in Breast Cancer: A Systematic Review and Meta-Analysis of Ultrasound, MRI and FDG PET

**DOI:** 10.3390/jcm10071543

**Published:** 2021-04-06

**Authors:** Morwenn Le Boulc’h, Julia Gilhodes, Zara Steinmeyer, Sébastien Molière, Carole Mathelin

**Affiliations:** 1Department of Oncologic Radiology, Claudius Regaud Institute, Institut Universitaire du Cancer de Toulouse-Oncopole, 31100 Toulouse, France; morwenn.le.boulch@gmail.com; 2Clinical Trials, Institut Universitaire du Cancer de Toulouse-Oncopole, 31100 Toulouse, France; Gilhodes.julia@iuct-oncopole.fr; 3Internal Medicine and Oncogeriatry Unit, Geriatric Department, University Hospital, Place du Docteur Baylac, CEDEX 9, 31059 Toulouse, France; Steinmeyer.Z@chu-toulouse.fr; 4Department of Women’s Imaging, University Hospitals of Strasbourg, 67200 Strasbourg, France; Sebastien.moliere@chru-Strasbourg.fr; 5Surgery at ICANS Cancer Institute (Institute of Cancerology Strasbourg Europe), CEDEX, 67033 Strasbourg, France

**Keywords:** meta-analysis, ultrasound, magnetic resonance imaging, positron emission tomography, breast cancer, lymph node, micrometastasis

## Abstract

Background: This systematic review aimed at comparing performances of ultrasonography (US), magnetic resonance imaging (MRI), and fluorodeoxyglucose positron emission tomography (PET) for axillary staging, with a focus on micro- or micrometastases. Methods: A search for relevant studies published between January 2002 and March 2018 was conducted in MEDLINE database. Study quality was assessed using the QUality Assessment of Diagnostic Accuracy Studies checklist. Sensitivity and specificity were meta-analyzed using a bivariate random effects approach; Results: Across 62 studies (*n* = 10,374 patients), sensitivity and specificity to detect metastatic ALN were, respectively, 51% (95% CI: 43–59%) and 100% (95% CI: 99–100%) for US, 83% (95% CI: 72–91%) and 85% (95% CI: 72–92%) for MRI, and 49% (95% CI: 39–59%) and 94% (95% CI: 91–96%) for PET. Interestingly, US detects a significant proportion of macrometastases (false negative rate was 0.28 (0.22, 0.34) for more than 2 metastatic ALN and 0.96 (0.86, 0.99) for micrometastases). In contrast, PET tends to detect a significant proportion of micrometastases (true positive rate = 0.41 (0.29, 0.54)). Data are not available for MRI. Conclusions: In comparison with MRI and PET Fluorodeoxyglucose (FDG), US is an effective technique for axillary triage, especially to detect high metastatic burden without upstaging majority of micrometastases.

## 1. Introduction

Breast cancer is the most commonly diagnosed cancer among women worldwide [1], accounting for 25% of cancer cases and 15% of cancer-related deaths [2]. Axillary lymph node (ALN) metastases are detected in 30 to 40% of women with breast cancer and are associated with a less favorable prognostic [3,4]. Sentinel lymph node biopsy (SLNB) is the classical staging procedure for breast cancer patients with clinically and radiologically negative axilla [5,6,7,8]. Preoperative detection of ALN involvement by imaging may change management in several ways, from first-line ALN dissection to neoadjuvant chemotherapy [9]. However, it is now well established that axillary micro- and macrometastases do not have the same prognostic and therapeutic impact, and the detection of micrometastasis should not lead to an ALN dissection or an inappropriate chemotherapy. Consequently, the axillary staging by imaging should help selecting patients with macrometastatic ALN and patients with negative or micrometastatic ALN.

To our knowledge, no study has systematically evaluated the performance of each of the 3 main imaging techniques as a triage test for axilla staging for breast cancer patients, especially without palpable ALN, with a focus on the type of nodal involvement (micro-or macrometastases). Many of the previous analyses concerning axillary staging did not include nodal ultrastadification and were performed in a population in which a significant proportion of patients had palpable ALN. Palpable ALN constitute a contraindication for SLNB as grossly involved nodes may not retain the dye or the radio-colloid agent due to the replacement of macrophages by cancer cells [10,11,12,13]. Moreover, inclusion of a significant proportion of patients undergoing neoadjuvant chemotherapy may not allow an accurate evaluation, as node staging may change during neoadjuvant chemotherapy (false negative).

Hence, the role and performance of imaging (including ultrastadification) remains to be clarified for breast cancer patients without palpable ALN, as well as the choice of the adequate imaging modality.

In clinical routine, axillary ultrasound (US) is widely performed, followed by fine-needle aspiration or core needle biopsy of abnormal ALN [3]. In some patients, magnetic resonance imaging (MRI) and 2-[18F]-fluoro-2-deoxy-D-glucose positron emission tomography (PET) are performed, for local or distant staging, and are potential techniques to improve axillary staging [9,14,15,16].

This systematic review aimed at systematically evaluating the performances of US (with or without fine-needle aspiration or core needle biopsy), MRI, and fluorodeoxyglucose PET for axillary staging, with a focus on micro- or micrometastases in breast cancer patients without palpable axillary nodes, and to discuss their use in different clinical settings.

## 2. Materials and Methods

### 2.1. Search Strategy

This systematic review followed the recommendations in the PRISMA statement [17,18]. Two reviewers independently searched the relevant studies that assessed the accuracy and the utility of US, MRI, and PET in staging the axilla in patients with breast cancer. The MEDLINE database was used for all in vivo human studies. The discrepancies were resolved by consensus.

### 2.2. Inclusion and Exclusion Criteria

Studies with the following inclusion criteria were reviewed: (1) Published in English, (2) cohort studies (prospective or retrospective); (3) published between 1 January 2002 and 15 March 2018; (4) imaging was done to detect ALN involvement in patients with breast cancer, (5) imaging procedures were US, MRI, PET; (6) histopathological analysis of ALN obtained by SLNB or ALN dissection procedure were used as the reference standard test, and (7) true positive (TP), false positive (FP), true negative (TN), and false negative (FN) values were reported or, if there was sufficient data for them, were calculated.

We excluded studies with the following criteria: (1) Neoadjuvant chemotherapy was administered between imaging and axillary surgery; (2) patients with palpable ALN ipsilateral to the breast cancer; (3) no histopathological reference standard; (4) patients without breast cancer; (5) insufficient data available to calculate the TP, FP, TN, and FN values; (6) imaging was performed for the sole purpose of detecting sentinel ALN; (7) patients were shared with another study previously included; (8) experimental subject was an animal and ex vivo; (9) under 18 analyzable patients in the study, (10) the type of study was a case control study, review, case report, letter to the editor, and (11) we were unable to get the full text.

Some studies were also included if we could manually exclude patients with exclusion criteria—such as patients treated with neoadjuvant chemotherapy or with palpable node, or patients without breast cancer and if we could calculate VP, FP, VN, and FN in the new population.

### 2.3. Data Extraction and Quality Assessment

Data were extracted by one reviewer, checked by a second, and discrepancies resolved by discussion. Study quality was assessed using the QUality Assessment of Diagnostic Accuracy Studies (QUADAS) checklist [19]. All the 14 items in the checklist were used.

### 2.4. Data Synthesis and Statistical Analysis

Patients were classified as TP when both imaging techniques and the reference standard (e.g., ALN dissection or SLNB) detected axillary metastases; TN when neither imaging techniques nor reference standard detected metastasis; FN when the imaging technique failed to detect metastasis identified by the reference standard; and FP when the imaging technique incorrectly suggested metastasis not detected by the reference standard. Sensitivity was defined as TP/(TP + FN) and SP as TN/(TN + FP). The diagnostic odd ratio (DOR) values was obtained with different combinations of SE and SP and could be used as a single summary measure. It was defined as the ratio of odds of positivity in disease relative to non-diseased. The DOR value ranges from 0 to infinity and a higher value means better diagnostic performance. A value of 1 indicates that a test cannot distinguish between patients with or without the disease and values of <1 introduce more FN results among the diseased [20].
(1)$$DOR\;=\;\left(TP/FN\right)/\left(FP/TN\right)=\;sens/\left(1-sens\right)/\left(1-spec\right)/spec$$

Considering the correlation between sensitivity and specificity, a bivariate random effects model was used to summarize performance estimates and their 95% confidence intervals (CI) [21]. Heterogeneity was assessed using the quantity I2 that lies between 0 and 100% (a value of 0% indicates no observed heterogeneity, values lower than 50% were considered as an acceptable level of heterogeneity) [22]. When no significant heterogeneity was observed between studies or when the number of considered studies was too small, a pooled analysis was undertaken. For all statistical tests, differences were considered significant at the 0.05 level. All statistical analyses were conducted using STATA 13.0^®^ software (copyright College Station, TX: StataCorp LP).

Forest plots were generated within Review Manager 5^®^ (copyright The Cochrane Collaboration, Copenhagen: The Nordic Cochrane Centre).

### 2.5. Subgroup Analyses

Subgroup analyses were undertaken according to US technique; US grayscale, US + fine needle aspiration/core needle biopsy, fine needle aspiration, and elastosonography. Subgroup analyses were conducted according to which MRI technique was used; MRI without diffusion weighted imaging (DWI), MRI with DWI, and DWI alone. Subgroup analyses were conducted according to which PET technique was used; PET without computed tomography (CT), and PET with CT.

In some studies, several results for one imaging technique, like MRI, were available, for example, for each MRI subgroup (e.g., MRI without DWI, MRI with DWI, DWI alone). As these results came from the same population, only one result could be considered for the pool estimates. Additionally, the subgroup with the best accuracy result ((TP + TN)/(TP + FP + FN + TN)) was considered.

For US studies, the US + fine needle aspiration/core needle biopsy criterion was preferred over US grayscale, because in routine clinical practice, any suspicious ALN in breast cancer undergoes ultrasound guided fine needle aspiration ore core needle biopsy. In studies evaluating elastosonography, nodes were considered abnormal if either US grayscale, elastosonography, or both were abnormal (disjunctive method).

Subgroups analysis were undertaken according to ALN involvement (micrometastases versus macrometastases and less than 3 ALN metastases versus 3 or more ALN metastases) in patient with T1–T2 breast cancer.

## 3. Results

### 3.1. Number and Characteristics of Included Studies

The search identified 569 citations from the MEDLINE data base, 95 were examined for full text review analysis after primary screening of titles and abstracts. Study characteristics of each subgroup are described in Table 1A–D.

In total, 62 studies were suitable for inclusion (Figure 1). There were 30 studies assessing US with or without fine needle aspiration/core needle biopsy, including 7546 patients of which 2668 had ALN metastases (prevalence = 35.4%), 10 studies assessing MRI, including 652 patients of which 211 had ALN metastases (prevalence = 32.4%), and 24 studies assessing PET, including 2388 patients of which 909 had ALN metastases (prevalence = 38.1%).

### 3.2. Quality of Included Studies

Figure 2 summarizes the methodological quality of the 62 included studies.

In general, the reference standard was adequate, but was not the same for all patients (either SLNB or ALN dissection), and the choice of the reference standard depended on the index test results (for instance, ALN dissection was performed for biopsy-proven metastatic nodes). The reference standard and the index test were well described in every study.

The index test was interpreted by reviewers blinded to reference standard results in all studies. The index test was often interpreted by reviewers blinded to other clinical data, most of the cases for MRI and PET studies, but rarely in US studies. Uninterpretable results were discussed in only 5 studies.

### 3.3. Sensitivity and Specificity of US, MRI and PET

Of the 30 studies evaluating US, sensitivity was 55% (95% CI: 49–62%; range 24–94%) and specificity was 99% (95% CI: 97–100%; range 76–100%). Of the 10 studies evaluating MRI, sensitivity was 83% (95% CI: 72–91%; range 50–100%) and specificity was 85% (95% CI: 72–92%; range 44–100%). Of the 24 studies evaluating PET, sensitivity was 49% (95% CI: 39–59%; range 19–84%) and specificity was 94% (95% CI: 91–96%; range 74–100%).

Results are presented in Table 2 and Figure 3A–C.

### 3.4. US Subgroups Analysis

Of 24 studies evaluating US grayscale only (*N* = 5575, prevalence: 37.4%), sensitivity was 63% (95% CI: 56–69%; range 28–88%) and specificity was 88% (95% CI: 82–92%; range 38–100%). Of 20 studies evaluating US + fine needle aspiration/core needle biopsy (*N* = 4874, prevalence: 33.1%), sensitivity was 51% (95% CI: 43–59%; range 24–94%) and specificity was 100% (95% CI: 99–100%; range 89–100%). Across 14 studies evaluating fine needle aspiration (*N* = 2404 patients, prevalence: 42.1%), sensitivity was 78% (95% CI: 73–83%; range 47–90%) and specificity was 99% (95% CI: 96–100%; range 91–100%). Only 2 studies evaluated elastosonography, not allowing meta-analysis: They both demonstrated a better sensitivity for US + elastosonography (disjunctive method) than elastosonography alone, but a lesser specificity. Results are presented in Table 2.

### 3.5. MRI Subgroups Analysis

Of the 7 studies evaluating MRI without DWI (*N* = 375, prevalence: 35.2%), sensitivity was 81% (95% CI: 49–95%; range 24–82%) and specificity was 84% (95% CI: 74–91%; range 54–100%). Of the 4 studies evaluating MRI with DWI (*N* = 366, prevalence: 31.4%), sensitivity was 78% (95% CI: 60–89%; range 54–95%) and specificity was 90% (95% CI: 82–95%; range 84–97%). Of the 5 studies evaluating DWI only (*N* = 398, prevalence: 32.9%), sensitivity was 74% (95% CI: 50–89%; range 40–83%) and specificity was 78% (95% CI: 51–92%; range 44–100%). Results are presented in Table 2.

### 3.6. PET Subgroups Analysis

Of the 9 studies evaluating PET without CT (*N* = 553, prevalence: 48.3%), sensitivity was 44% (95% CI: 28–62%; range 20–84%) and specificity was 95% (95% CI: 91–97%; range 85–100%). Of the 15 studies evaluating PET with CT (*N* = 1835, prevalence: 35%), sensitivity was 51% (95% CI: 40–63%; range 19–81%) and specificity was 93% (95% CI: 89–96%; range 74–100%). Results are presented in Table 2.

### 3.7. Subgroup Analysis on Axillary Metastatic Burden

In 12 studies (1497 patients), data about axillary burden were presented, including the histological size of the largest ALN metastasis. The overall preoperative FN rate was 0.93 (0.87, 0.97) for micrometastasis and 0.56 (0.51, 0.61) for macrometastasis. For US (705 patients), the FN rate was 0.96 (0.86, 0.99) for micrometastasis, and 0.52 (0.45, 0.59) for macrometastasis. For PET (643 patients), the FN was 0.59 (0.46, 0.71) for micrometastasis, and 0.64 (0.56, 0.71) for macrometastasis. No subgroup analysis was possible for MRI due to the lack of data.

The number of involved ALN in early-stage breast cancer patients (T1 or T2) was given in 4 studies. For ultrasonography (632 patients), the FN rate was 0.63 (0.57, 0.68) for 1 or 2 involved node(s) and 0.28 (0.22, 0.34) when 3 or more nodes were involved.

## 4. Discussion

In this meta-analysis assessing the diagnostic performances of US, MRI, and PET for pretherapeutic ALN staging, we found that while MRI had a significant higher sensitivity than other imaging modalities, the performance of US significantly improved for macrometastases in more than 2 ALN. The association of US and fine needle aspiration had the highest diagnostic odd ratio, in part because of a specificity close to 100%.

Unlike other published meta-analysis, we chose to assess each of these 3 techniques to put in contrast their respective strengths and weaknesses and to offer an overview of the role of imaging for nodal staging and ultrastadification.

We did not include patients with clinically positive ALN, for which preoperative imaging is unlikely to change treatment plan [12]. We also chose not to include patients undergoing neoadjuvant chemotherapy, in order to have a gold-standard reference test available for every patient.

While previously published meta-analysis had a high prevalence of ALN metastasis [3,11], the metastasis rate in our study was in line with the commonly described rate of ALN metastasis in invasive breast cancer, between 30 and 40% [3,4].

Management of axilla has evolved with the increased use of neoadjuvant treatment. Furthermore, the ACOSOG Z0011 trial proved that women with micrometastases or less than 2 metastatic ALN and clinical T1-2 tumors undergoing lumpectomy and breast radiation therapy followed by systemic therapy, did not benefit from ALN dissection in terms of local control and 10-year overall survival [13]. An ideal preoperative axillary staging should therefore be able not only to detect macrometastasis with high accuracy, but also to evaluate the global axillary burden, in order to avoid unnecessary ALN dissection in low axillary burden.

We found that axillary US has a very high specificity (99%, 95% CI: 97–100%), in contrast with its much lower overall sensitivity [85,86], which indeed depends on the axillary burden: FN rate of US drops to 0.28 when more than 2 ALN are involved, while micrometastases are almost never detected. This data is fundamental to avoid over-treatment, as micrometastasis should not lead to an ALN dissection or the prescription of chemotherapy. A recent study on interobserver variability showed that the discrimination between low and high axillary burden on US is reliable and reproducible [87]. US should be used for first-line axillary triage, to detect high metastatic burden that could benefit from neoadjuvant chemotherapy, without diverting low-burden patients from SLNB procedure. Technical improvements, such as elastosonography [23,25] or the use of intradermal microbubbles to locate and biopsy the sentinel lymph node under ultrasound guidance [88] may further increase US sensitivity.

We found that MRI has a better sensitivity than US for detection of nodal metastasis. This is in line with the results of other meta-analysis, for example, Liang et al. [7] found a sensitivity of 82% (95%CI: 78–86). The main drawback of MRI is its relatively low specificity compared to other imaging modalities, which makes it unsuitable for surgical or oncological planning. The adjunction of diffusion-weighting imaging seems to significantly increase its specificity while only slightly decreasing its sensitivity. In one study by Hieken et al. [89], second-look US after abnormal axillary findings on MRI allowed detection of abnormal nodes not previously detected by US in only 10% of the cases. In the clinical situation of a positive MRI with negative US, there is a significant risk of axillary false positive.

In our study, PET shows a lower sensitivity than in Cooper’s less recent meta-analysis (49% vs. 63%) [4]. Indeed, performance of PET may vary depending on breast cancer histological subtypes, with higher performances in basal than luminal subtypes [90,91] and also depending on the histological gold standard (e.g., high rate of micrometastases in recent studies [15]). A functional, high-sensitivity imaging, PET has a much higher detection rate of micrometastases than US, which can theoretically lead to unnecessary ALN dissection or neoadjuvant chemotherapy. Yet, PET has the unique ability to detect extranodal distant metastasis and should be used preferentially in patients at high risk for extranodal disease. Further technical improvements, especially new markers for hormone-positive or HER2-positive breast tumors, may redefine the role of PET imaging in axillary staging.

Our study has some limits. A relatively low number of MRI studies were included in our metanalysis, as this imaging modality has only been studied more recently for axilla staging. Likewise, probably due to the lower availability of MRI and PET, these modalities are more widely used for T3-T4 than T1-T2 stages. It may explain why MRI and PET studies include fewer T1-T2 breast cancer than US studies. However, the prevalence of ALN metastases for each of 3 modalities was roughly the same, between 30 and 40%. High heterogeneity of MRI subgroup analysis was probably due to the lack of consensus on the criteria used to define a suspicious ALN on MRI, as well as difference in imaging protocol between centers (MRI field strength, imaging parameters). Finally, information about axillary burden was not widely available in MRI and PET studies.

Thus, future imaging studies should systematically include such parameters as the number of metastatic ALN, the presence of micrometastases versus macrometastases, and the presence of a capsular rupture to avoid over diagnosis and over treatment.

## 5. Conclusions

US is an effective technique for axillary triage, especially to detect high metastatic burden that could benefit from neoadjuvant chemotherapy or axillary clearance, without upstaging the majority of micrometastases.

## Figures and Tables

**Figure 1 jcm-10-01543-f001:**
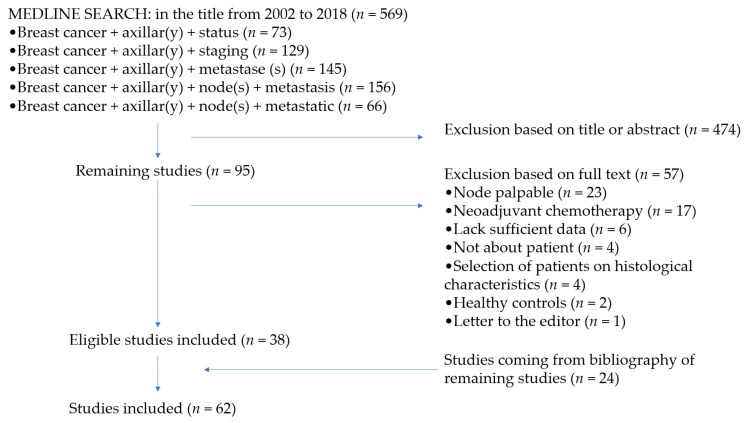
Flowchart depicting the inclusion and exclusion of the identified studies.

**Figure 2 jcm-10-01543-f002:**
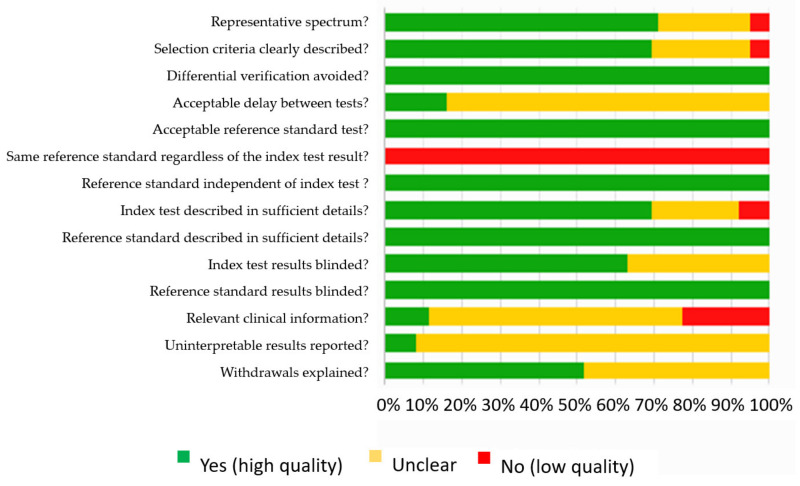
Quality analysis of the included studies based on QUality Assessment of Diagnostic Accuracy Studies (QUADAS).

**Figure 3 jcm-10-01543-f003:**
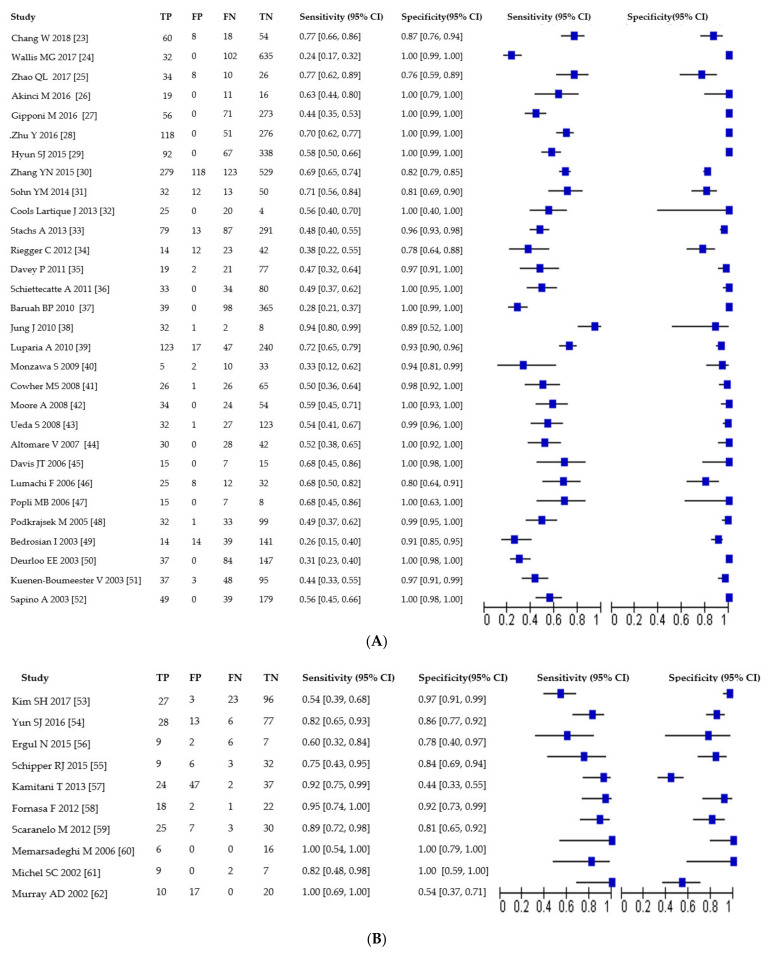
(**A**) Forest plot of sensitivity and specificity for US studies. TP = true positive, FP = false positive, FN = false negative, TN = true negative. Brackets show 95% confidence intervals. The figure shows the sensitivity and specificity for each study (squares) and 95% confidence intervals (horizontal lines). (**B**) Forest plot of sensitivity and specificity for MRI studies. TP = true positive, FP = false positive, FN = false negative, TN = true negative. Brackets show 95% confidence intervals. The figure shows the sensitivity and specificity for each study (squares) and 95% confidence intervals (horizontal lines). (**C**) Forest plot of sensitivity and specificity for PET studies. TP = true positive, FP = false positive, FN = false negative, TN = true negative. Brackets show 95% confidence intervals. The figure shows the sensitivity and specificity for each study (squares) and 95% confidence intervals (horizontal lines).

**Table 1 jcm-10-01543-t001:** Study characteristics.

(**A**) Characteristics of ultrasound included studies													
**Author**	**Year**	**Country**	**Index Test**	**Second Test**	**Reference Standard**		**Prospective/Retrospective**	**N Analysed**	**N with Axillary Metastases**	**Prevalence of Axillary Metastases**	**Mean Age**	**Years of Study**	**Other Criteria**
Chang W. [23]	2018	China	US	US + Elastosonography	Histology (SLNB/ALND)		Retrospective	140	78	55.7%	55.3	2013–2014	Disjunctive method
Wallis M.G. [24]	2017	UK	US	US + CNB	Histology (SLNB/ALND)		Retrospective	769	134	17.4%	ND	2008–2015	
Zhao Q.L. [25]	2017	China	US	US + Elastosonography	Histology (SLNB/ALND)		Prospective	78	44	56.4%	52.5	2012–2013	Disjunctive method
Akinci M. [26]	2016	Turkey	US	US + FNA	Histology (SLNB/ALND)		Prospective	46	30	65.2%	ND	2011–2013	
Gipponi M. [27]	2016	Italy	US	US + FNA	Histology (SLNB/ALND)		Prospective	400	127	31.8%	64.6	2013–2015	Only T1-T2-T3 tumors
Zhu Y. [28]	2016	China	US	US + FNA	Histology (SLNB/ALND)		Retrospective	445	169	38.0%	55.6	2013–2014	Only T1-T2 tumors
Hyun S.J. [29]	2015	South Korea	US	US + FNA	Histology (SLNB/ALND)		Retrospective	497	159	32.0%	52	2012–2013	
Zhang Y.N. [30]	2015	China	US	US grayscale	Histology (SLNB/ALND)		Retrospective	1049	402	38.3%	50.3	2010–2011	
Sohn Y.M.b[31]	2014	South Korea	US	US grayscale	Histology (SLNB/ALND)		Retrospective	107	45	42.1%	53.9	2009–2012	
Cools Lartique J. [32]	2013	Canada	US	US + FNA	Histology (SLNB/ALND)		Prospective	234	90	38.5%	57.8	2005–2007	
Stachs A. [33]	2013	Germany	US	US grayscale	Histology (SLNB/ALND)		Retrospective	470	166	35.3%	ND	2008–2010	
Riegger C. [34]	2012	Germany	US	US grayscale	Histology (SLNB/ALND)		Retrospective	91	37	40.7%	55.5	2007–2010	
Davey P. [35]	2011	Northern Ireland	US	US + FNA	Histology (SLNB/ALND)		Retrospective	119	40	33.6%	ND	2009	
Schiettecatte A. [36]	2011	Belgium	US	US + FNA	Histology (SLNB/ALND)		Retrospective	147	67	45.6%	56	ND	Breast tumors < 3cm
Baruah B.P. [37]	2010	UK	US	US + FNA	Histology (SLNB/ALND)		Retrospective	502	137	27.3%	61	2006–2009	
Jung J. [38]	2010	South Korea	US	US + FNA	Histology (SLNB/ALND)		Retrospective	189	61	32.3%	ND	2005–2006	
Luparia A. [39]	2010	Italy	US	US grayscale	Histology (SLNB/ALND)		Retrospective	427	170	39.8%	60.9	2005–2008	
Monzawa S. [40]	2009	Japan	US	US grayscale	Histology (SLNB/ALND)		Retrospective	50	15	30.0%	59	2005–2006	
Cowher M.S. [41]	2008	USA	US	US + FNA	Histology (SLNB/ALND)		Retrospective	125	57	45.6%	61.3	2004–2005	
Moore A. [42]	2008	USA	US	US + FNA	Histology (SLNB/ALND)		Retrospective	112	58	51.8%	ND	ND	High risk of metastases
Ueda S. [43]	2008	Japan	US	US grayscale	Histology (SLNB/ALND)		Prospective	183	59	32.2%	57	2005–2007	
Altomare V. [44]	2007	Italy	US	US + FNA	Histology (SLNB/ALND)		Retrospective	100	30	30.0%	53	2004–2005	Only T1-T2-T3 tumors. FNA performed for all patients
Davis J.T. [45]	2006	USA	US	US + FNA	Histology (SLNB/ALND)		Prospective	37	22	59.5%	ND	2004–2005	High risk of metastases
Lumachi F. [46]	2006	Italy	US	US grayscale	Histology (SLNB/ALND)		Prospective	77	37	48.1%	54	ND	Only T1-T2 tumors.
Popli M.B. [47]	2006	India	US	US + FNA	Histology (SLNB/ALND)		Prospective	30	22	73.3%	ND	ND	
Podkrajsek M. [48]	2005	Slovenia	US	US + FNA	Histology (SLNB/ALND)		Retrospective	165	65	39.4%	56	2001–2003	
Bedrosian I. [49]	2003	USA	US	US + FNA	Histology (SLNB/ALND)		Prospective	208	53	25.5%	55.4	1994–2000	
Deurloo E.E. [50]	2003	The Netherlands	US	US + FNA	Histology (SLNB/ALND)		Prospective	268	121	45.1%	56	1999–2001	Only patients eligible for SLNB
Kuenen-Boumeester V. [51]	2003	The Netherlands	US	US + FNA	Histology (SLNB/ALND)		Retrospective	183	85	46.4%	ND	1998–2003	
Sapino A. [52]	2003	Italy	US	US + FNA	Histology (SLNB/ALND)		Prospective	298	88	29.5%	ND	2000	31 in situ breast cancer
TOTAL								7546	2668	35.4%	56		
(**B**) Characteristics of Magnetic Resonance Imaging included studies													
**Author**	**Year**	**Country**	**Index Test**	**Second Test**	**Number of Testla**	**Reference Standard**	**Prospective/Retrospective**	**N Analysed**	**N with Axillary Metastases**	**Prevalence of Axillary Metastases**	**MEAN AGE**	**Period of Study**	**Other Criteria**
Kim S.H. [53]	2017	South Korea	MRI	With and without DWI + Gadolinium IV	3T	Histology	Retrospective	149	50	33.6%	49.2	2014–2015	
						(SLNB/ALND)							
Yun S.J. [54]	2016	South Korea	MRI	With DWI + Gadolinium IV	3T	Histology	Retrospective	124	34	27.4%	59.8	2011–2014	
						(SLNB/ALND)							
Schipper R.J. [55]	2015	The Netherlands	MRI	With and without DWI	3T	Histology	Retrospective	50	12	24.0%	60	2012–2013	Only T1-T2-T3
						(SLNB/ALND)							tumors
Ergul N. [56]	2015	Turkey	MRI	With and without DWI	1.5T	Histology	Prospective	24	15	62.5%	47	2012–2013	Only T1-T2
						(SLNB/ALND)							tumors
Kamitani T. [57]	2013	Japan	MRI	DWI alone	1.5T	Histology	Retrospective	110	26	23.6%	54.9	2006–2007	
						(SLNB/ALND)							
Fornasa F. [58]	2012	Italy	MRI	With DWI + Gadolinium IV	1.5T	Histology	Prospective	43	19	44.2%	58	2008–2010	
						(SLNB/ALND)							
Scaranelo M. [59]	2012	Canada	MRI	With and without DWI	1.5T	Histology	Prospective	65	28	43.1%	53	2008–2009	
						(SLNB/ALND)							
Memarsadeghi M. [60]	2006	Austria	MRI	Without DWI + USPIO IV	1T	Histology	Prospective	22	6	27.3%	62	5 months	
						(SLNB/ALND)							
Michel S.C. [61]	2002	Switzerland	MRI	Without DWI + USPIO IV	1.5T	Histology	Prospective	18	11	61.1%	53	2000–2001	
						(SLNB/ALND)							
Murray A.D. [62]	2002	UK	MRI	Without DWI + Gadolinium IV	0.95T	Histology	ND	47	10	21.3%	63	ND	
						(SLNB/ALND)							
TOTAL								652	211	32.4%	55.4		
(**C**) Characteristics of FDG Positron Emission Tomography included studies													
**Author**	**Year**	**Country**	**Index Test**	**Second Test**	**Evaluation**	**Reference Standard**	**Prospective/Retrospective**	**N Analysed**	**N with Axillary Metastases**	**Prevalence of Axillary Metastases**	**Mean Age**	**Years of Study**	**Other Criteria**
Ergul N. [56]	2015	Turkey	FDG PET	With CT	Visual and semi-quantitative	Histology	Prospective	24	15	62.5%	47	2012–2013	Only T1-T2 tumors
						(SLNB/ALND)							
Jeong Y.J. [63]	2014	South Korea	FDG PET	With CT	Visual and semi-quantitative	Histology	Retrospective	178	48	27.0%	54.9	2010–2013	
						(SLNB/ALND)							
Park J. [64]	2014	South Korea	FDG PET	With CT	Visual and semi-quantitative	Histology	Retrospective	136	70	51.5%	49.7	2009–2012	3 patients without FDG-avid breast tumors excluded
						(SLNB/ALND)							
Sohn Y.M. [31]	2014	South Korea	FDG PET	With CT	Visual	Histology	Retrospective	107	45	42.1%	53.9	2009–2012	
						(SLNB/ALND)							
Machida Y. [65]	2013	Japan	FDG PET	With CT	Visual and semi-quantitative	Histology	Retrospective	227	54	23.8%	ND	2005–2009	
						(SLNB/ALND)							
Seok J.W. [66]	2013	South Korea	FDG PET	With CT	Visual and semi-quantitative	Histology	Retrospective	104	21	20.2%	49.4	2010–2012	Only T1-T2 tumors
						(SLNB/ALND)							
Hahn S. [67]	2012	Germany	FDG PET	With CT	Visual and semi-quantitative	Histology	Retrospective	38	16	26.9%	52	2008	Only T1-T2 tumors
						(SLNB/ALND)							
Riegger C. [34]	2012	Germany	FDG PET	With CT	Visual	Histology	Retrospective	91	37	40.7%	55.5	2007–2010	
						(SLNB/ALND)							
Choi W.H. [68]	2011	South Korea	FDG PET	With CT	Visual and semi-quantitative	Histology	Retrospective	171	73	42.7%	50.1	2003–2006	
						(SLNB/ALND)							
Heusner T.A. [69]	2009	Germany	FDG PET	With CT	Visual	Histology	Retrospective	61	24	39.3%	56	2007–2008	
						(SLNB/ALND)							
Kim J [70]	2009	South Korea	FDG PET	With CT	Visual	Histology	Prospective	137	35	25.5%	50.5	2007–2008	Only T1-T2 tumors
						(SLNB/ALND)							
Monzawa S. [40]	2009	Japan	FDG PET	With CT	Visual	Histology	Retrospective	50	15	30.0%	59	2005–2006	
						(SLNB/ALND)							
Taira N. [71]	2008	Japan	FDG PET	With CT	Visual and semi-quantitative	Histology	Retrospective	92	27	29.3%	54.6	2006–2007	
						(SLNB/ALND)							
Ueda S. [43]	2008	Japan	FDG PET	With CT	Visual	Histology	Prospective	183	59	32.2%	57	2005–2007	
						(SLNB/ALND)							
Veronesi U. [72]	2007	Italy	FDG PET	With CT	Visual and semi-quantitative	Histology	Retrospective	236	103	43.6%	49	2003–2005	Only T1-T2-T3 tumors
						(SLNB/ALND)							
Kumar R. [73]	2006	USA	FDG PET	Without CT	ND	Histology	Prospective	80	36	45.0%	52	ND	
						(SLNB/ALND)							
Weir L. [74]	2005	Canada	FDG PET	Without CT	Visual	Histology	Retrospective	40	18	45.0%	52	2000–2003	
						(SLNB/ALND)							
Fehr M.K. [75]	2004	Switzerland	FDG PET	Without CT	Visual	Histology	Prospective	24	10	41.7%	56	ND	Tumors
						(SLNB/ALND)							< 3 cm (clinical)
Zornoza M.J. [76]	2004	Spain	FDG PET	Without CT	Visual	Histology	Prospective	200	107	53.5%	52.2	ND	Tumors < 3.5 cm (ND)
						(SLNB/ALND)							
Barranger E. [77]	2003	France	FDG PET	Without CT	Visual	Histology	Prospective	32	15	46.9%	58	2001	Only T1-T2 tumors
						(SLNB/ALND)							
Guller U. [78]	2002	Switzerland	FDG PET	Without CT	ND	Histology	Prospective	31	14	45.2%	64.8	ND	
						(SLNB/ALND)							
Nakamoto Y. [79]	2002	USA	FDG PET	Without CT	Visual	Histology	Prospective	36	15	41.7%	50.6	ND	
						(SLNB/ALND)							
Rieber A. [80]	2002	Germany	FDG PET	Without CT	ND	Histology	Retrospective	40	20	50.0%	52.9	ND	
						(SLNB/ALND)							
Van der Hoeven J.M. [81]	2002	The Netherlands	FDG PET	Without CT	Visual	Histology (SLNB/ALND)	Prospective	70	32	45.7%	58	1997–2000	
TOTAL								2388	909	38.1%	52.9		
(**D**) Characteristics of Fine Needle Aspiration included studies													
**Author**	**Year**	**Country**	**Index Test**	**Evaluation**	**Prospective/Retrospective?**	**N Analysed**	**N with Axillary Metastases**	**Prevalence of Axillary Metastases**	**Mean Age**	**Years of Studies**	**Other Criteria**		
Zhu Y. [28]	2016	China	FNA	Histology	Retrospective	445	169	38.0%	55.6	2013–2014	Only T1-T2 tumors		
				(SLNB/ALND)									
Sohn Y.M. [31]	2014	South Korea	FNA	Histology	Retrospective	107	45	42.1%	53.9	2009–2012			
				(SLNB/ALND)									
Ganott M.A. [82]	2014	USA	FNA	Histology	Prospective	44	26	59.1%	ND	2008–2010			
				(SLNB/ALND)									
Hayes B.D. [83]	2011	Ireland	FNA	Histology	Retrospective	161	86	53.4%	ND	2006–2009			
				(SLNB/ALND)									
Schiettecatte A. [36]	2011	Belgium	FNA	Histology	Retrospective	147	67	45.6%	56	ND			
				(SLNB/ALND)									
Luparia A. [39]	2010	Italy	FNA	Histology	Retrospective	427	170	39.8%	60.9	2005–2008	FNA was not performed for all suspicious axillary US		
				(SLNB/ALND)									
Tahir M. [84]	2008	UK	FNA	Histology	Prospective	38	17	44.7%	56.7	2005–2006			
				(SLNB/ALND)									
Cowher M.S. [41]	2008	USA	FNA	Histology	Retrospective	125	57	45.6%	61.3	2004–2005			
				(SLNB/ALND)									
Moore A. [42]	2008	USA	FNA	Histology	Retrospective	112	58	51.8%	ND	ND	Only high risk of metastases		
				(SLNB/ALND)									
Davis J.T. [45]	2006	USA	FNA	Histology	Prospective	37	22	59.5%	ND	2004–2005	Only high risk of metastases		
				(SLNB/ALND)									
Popli M.B. [47]	2006	India	FNA	Histology	Prospective	30	22	73.3%	ND	ND			
				(SLNB/ALND)									
Podkrajsek M. [48]	2005	Slovenia	FNA	Histology	Retrospective	165	65	39.4%	56	2001–2003			
				(SLNB/ALND)									
Deurloo E.E. [50]	2003	The Netherlands	FNA	Histology	Prospective	268	121	45.1%	56	1999–2001			
				(SLNB/ALND)									
Sapino A. [52]	2003	Italy	FNA	Histology	Prospective	298	88	29.5%	ND	2000			
				(SLNB/ALND)									
TOTAL						2404	1013	42.1%	49.9				

ALND: Axillary Lymph Nodes Dissection; SLNB: Sentinel Lymph Node Biopsy; CNB: Core Needle Biopsy; FNA: Fine Needle Aspiration; DWI: Diffusion Weighted Imaging; IV: Intravenous injection; MRI: Magnetic Resonance Imaging; CT: Computed Tomography; FDG: Fluorodeoxyglucose; PET: Positron Emission Tomography; USPIO: Ultrasmall Superparamagnetic Iron Oxide; US: Ultrasonography; N: Number of patients; ND: Not Determined; UK: United Kingdom; USA: United States of America.

**Table 2 jcm-10-01543-t002:** Summary estimates of sensitivity, specificity, diagnostic odds ratio, and their 95% confidence intervals of US, MRI, and FDG PET.

Imaging Technique	N Studies	Sensitivity	I^2^	Specificity	I^2^	DOR
US	30	0.55 (0.49, 0.62)	90.01	0.99 (0.97, 1.00)	95.06	112 (39, 320)
US grayscale	24	0.63 (0.56, 0.69)	88.86	0.88 (0.82, 0.92)	93.91	12 (8, 18)
US + FNA|CNB	20	0.51 (0.43, 0.59)	88.44	1.00 (0.99, 1.00)	94.19	752 (98, 5765)
FNA	14	0.78 (0.73, 0.83)	55.40	0.99 (0.96, 1.00)	48.73	560 (91, 3451)
MRI	10	0.83 (0.72, 0.91)	75.81	0.85 (0.72, 0.92)	93.00	28 (16, 51)
MRI without DWI	7	0.81 (0.49, 0.95)	89.17	0.84 (0.74, 0.91)	89.04	22 (7, 72)
MRI with DWI	4	0.78 (0.60, 0.89)	79.35	0.90 (0.82, 0.95)	67.07	33 (17, 65)
DWI alone	5	0.74 (0.50, 0.89)	83.54	0.78 (0.51, 0.92)	93.63	10 (5, 19)
PET FDG	24	0.49 (0.39, 0.59)	87.03	0.94 (0.91, 0.96)	73.98	15 (8, 26)
PET FDG without CT	9	0.44 (0.28, 0.62)	90.90	0.95 (0.91, 0.97)	0	14 (5, 44)
PET FDG with CT	15	0.51 (0.40, 0.63)	86.04	0.93 (0.89, 0.96)	79.51	14 (8, 27)

CNB: Core Needle Biopsy; CT: Computed Tomography; DOR: Diagnostic Odds Ratio; DWI: Diffusion Weighed Imaging; FDG: Fluorodeoxyglucose; FNA: Fine Needle Aspiration; MRI: Magnetic Resonance Imaging; PET: Positron Emission Tomography; US: Ultrasonography. The diagnostic odd ratio (DOR) values obtained with different combinations of sensitivity and specificity could be used as a single summary measure. It was defined as the ratio of odds of positivity in disease relative to non-diseased. The DOR value ranges from 0 to infinity, and a higher value signifies better diagnostic performance. A value of 1 indicates that a test cannot distinguish between patients with or without the disease and values of <1 introduce more FN results among the diseased [22]. Confidence intervals consider the heterogeneity beyond chance between studies (random effects models). The impact of unobserved heterogeneity is traditionally assessed statistically using the quantity I2. It describes the percentage of total variation across studies that is attributable to the heterogeneity rather than chance [22]. Magnetic resonance imaging (MRI) had a significantly higher sensitivity than other imaging modalities, whereas Ultrasonography (US) had a significantly higher specificity than MRI and to a lesser extent than fluorodeoxyglucose positron emission tomography (PET). DOR estimated for US was significantly greater than those of MRI, which in turn was significantly greater than those of FDG PET. Further analysis revealed that for all imaging modality, US + fine needle aspiration (FNA) or core needle biopsy (CNB) had the highest DOR value. For MRI studies, MRI with diffusion weighted imaging (DWI) had the highest DOR value and for PET studies. PET with or without computed tomography (CT) had the same DOR value.

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
