# Peer review of "Pretherapeutic Imaging for Axillary Staging in Breast Cancer: A Systematic Review and Meta-Analysis of Ultrasound, MRI and FDG PET"

_jcm, 2021, doi:10.3390/jcm10071543_

Round 1

Reviewer 1 Report

Accurasy of Ultrasound is very much operator dependant, any comments on how reprodusible will this results be and how mcuh it will depend on the experience of the person doing the US

Author Response

Accuracy of Ultrasound is very much operator dependent, any comments on how reproducible these results will be and how much it will depend on the experience of the person doing the US

Thank you for your review and your comments.

Indeed, there are usually concerns about the inter-reader variability of ultrasound studies in general, in breast diseases in particular. Despite the lack of published data on this particular topic, it seems that differentiation between low axillary burden and high axillary burden is subjected to relatively low inter-reader variability. We added contents in the discussion and a reference about that important issue.

Reviewer 2 Report

good review and good discussion.

Maybe discuss US and MR, is the combination better?

Author Response

REVIEWER 2

good review and good discussion.

Maybe discuss US and MR, is the combination better?

Thank you for your review and for raising this important point.

It is routine practice to do second-look axillary US after a positive axilla on MRI. This second-look US only detect a small number of abnormal nodes not previously detected but allows cytological documentation of these cases. In case second-look US is negative despite a positive MRI, the axillary burden is probably low. We added contents in the discussion and a reference about this clinical scenario.

Reviewer 3 Report

The comparison between the different diagnostic techniques I think is not assessable given the variability of the different studies (there are no studies that compare the three). Furthermore, as the authors describe, there are already similar meta-analyzes. The discussion focuses on the authors' preconceptions not on the results of the systematic review.

Author Response

REVIEWER 3

The comparison between the different diagnostic techniques I think is not assessable given the variability of the different studies (there are no studies that compare the three). Furthermore, as the authors describe, there are already similar meta-analyzes. The discussion focuses on the authors' preconceptions not on the results of the systematic review.

Thank you for your review and your comments.

Indeed, no study comparing the three imaging methods together is yet available. And yet despite of - and also because of - the lack of comparative data, we think our meta-analysis can offer interesting perspectives about the strength and weaknesses of each modality.

One major characteristic of our review, in contrast with existing reviews, is the attention we brought to only include studies focusing on a clinical problem in a specific population: which is the more appropriate imaging method to assess a clinically-negative axilla, before any treatment, in light of the change of surgical paradigm in a post-Z0011 era.

We modified the title and some portion of the manuscript to avoid misleading the reader about the notion of “comparison”.

Thanks to your comment on the discussion, we also significantly modified this section, beginning each subsection with a summary of the results of the review and subsequently developing the clinical relevance of these results, apart from any preconception but rather based on recent clinical paradigms.

Reviewer 4 Report

Review Pretherapeutic imaging for axillary staging in breast cancer: a systematic review and meta-analysis comparing ultrasound, MRI and FDG PET

This meta-analysis aimed at comparing different imaging modalities for axillary staging of breast cancer. A total of 10,374 patients from 62 studies were included. The prevalence of lymph nodes metastases was comparable for US studies, MRI studies and FDG PET studies, reaching about 30-40%. Authors focussed on discrimination between micro- and macrometastases as well as number of involved lymph node, as this is very important to avoid surgical overtreatment. They found that ultrasound was best suited to detect relevant axillary tumor burden.

I have one major concern:

The title oft he heading implies a true comparison between US, MRI and FDG PET. In this meta-analysis, however, in the included studies only one of these imaging methods was examined each. Therefore I would suggest to change the title of the heading by avoiding the word „comparing“.

Minor revisions:

Abstract:

Line 29: Please write MRI instead of RMI.

Author Response

REVIEWER 4

This meta-analysis aimed at comparing different imaging modalities for axillary staging of breast cancer. A total of 10,374 patients from 62 studies were included. The prevalence of lymph nodes metastases was comparable for US studies, MRI studies and FDG PET studies, reaching about 30-40%. Authors focused on discrimination between micro- and macrometastases as well as number of involved lymph node, as this is very important to avoid surgical overtreatment. They found that ultrasound was best suited to detect relevant axillary tumor burden.

I have one major concern:

The title of the heading implies a true comparison between US, MRI and FDG PET. In this meta-analysis, however, in the included studies only one of these imaging methods was examined each. Therefore I would suggest to change the title of the heading by avoiding the word „comparing“.

Minor revisions:

Abstract:

Line 29: Please write MRI instead of RMI.

Thank you for your review and your comments.

It is absolutely true, no comprehensive study comparing the three imaging methods together was available for our meta-analysis. This manuscript is in part the result of the relative lack of comparative data:  we included studies sharing common clinical characteristics (non-palpable axillary node, excluding neoadjuvant chemotherapy) and a similar prevalence of lymph node invasion (from 32.4 for MRI studies to 38.1% for PET studies) to offer perspectives about the strength and weaknesses of each modality in a common clinical situation.

Following your recommendation, we changed the title of the manuscript to avoid misleading the reader.